# Does Commercial Inoculation Promote Arbuscular Mycorrhizal Fungi Invasion?

**DOI:** 10.3390/microorganisms10020404

**Published:** 2022-02-09

**Authors:** Sulaimon Basiru, Mohamed Hijri

**Affiliations:** 1African Genome Center, Mohammed VI Polytechnic University (UM6P), Lot 660, Hay Moulay Rachid, Ben Guerir 43150, Morocco; sulaimon.basiru@um6p.ma; 2Département de Sciences Biologiques, Institut de Recherche en Biologie Végétale, Université de Montréal, 4101 Sherbrooke Est, Montréal, QC H1X 2B2, Canada

**Keywords:** arbuscular mycorrhizal fungi, commercial inoculants, invasion, indigenous/resident microbiota, native propagules, ecosystem function, community ecology

## Abstract

Interventions with commercial inoculants have the potential to reduce the environmental footprint of agriculture, but their indiscriminate deployment has raised questions on the unintended consequences of microbial invasion. In the absence of explicit empirical reports on arbuscular mycorrhizal fungi (AMF) invasion, we examine the present framework used to define AMF invasion and offer perspectives on the steps needed to avoid the negative impacts of AMF invasion. Although commercial AMF isolates are potential invaders, invasions do not always constitute negative impacts on native community diversity and functions. Instead, the fates of the invading and resident communities are determined by ecological processes such as selection, drift, dispersal, and speciation. Nevertheless, we recommend strategies that reduce overdependence on introduced inoculants, such as adoption management practices that promote the diversity and richness of indigenous AMF communities, and the development of native propagules as a supplement to commercial AMF in applicable areas. Policies and regulations that monitor inoculant value chains from production to application must be put in place to check inoculant quality and composition, as well as the transport of inoculants between geographically distant regions.

## 1. Introduction

Increasing the genetic and functional diversity of soil microbiota is a sustainable strategy to improve the efficiency and resilience of the agricultural system. Arbuscular mycorrhizal fungi (AMF) are an essential component of soil microbiota that play various roles in plant and soil health. By establishing symbiosis with approximately 80% of terrestrial plants [1], AMF act as an extension of underground plant root networks, facilitating plant growth by contributing to nutrient acquisition and stress alleviation in exchange for plant photosynthates [2,3]. Commercial AMF inoculants are deployed as biofertilizers in the agricultural field to enhance not only crop growth, but also the diversity and functions of indigenous AMF communities in nutrient-poor and degraded soils. However, the global distribution of microbial inoculants could constitute deliberate action toward microbial invasion [4] by disrupting plant holobionts and ecosystem services or acting as plant pathogens [5].

Although there are no convincing empirical data on AMF invasion to date, some studies have shown that introducing alien strains—isolates from different geographical ranges, or in vitro propagated strains that may be functionally divergent from their natural relatives [6]—into new agroecosystems can have far-reaching consequences on resident communities. The introduction of exotic AMF species into the field can lead to the stimulation, suppression, or exclusion of native microbial communities [7,8]. The potential negative consequences of AMF invasion on soil and plant biodiversity, as well as ecosystem functions, have also been raised [9]. Therefore, in this perspective, we view microbial invasion through the lens of the ecological frameworks proposed by Kinnunen et al., 2016 [10] and the community ecology concept described by Vellend (2010) [11]. Through these frameworks, we ask whether the deployment of commercial inoculants promotes AMF invasion, and whether successful invasion always generates negative impacts. We further provide suggestions on effective management practices and strategies that can help reduce the negative consequences associated with the deployment of commercial AMF.

## 2. Ecologic Framework for AMF Invasion

Microbial invasion occurs when microorganisms spread and proliferate in a new range (a place where they have never existed) and negatively impact the local community [12]. Thus, determining the invasiveness of a species or strain requires past and present knowledge of the community, whether the species or isolate has dispersed beyond its geographical range (or was never part of the present community), and whether there are negative impacts. This is, however, challenging due to the complex biogeography of AMF that encompasses cosmopolitan distribution [13,14], moderate endemism [15], and local adaptation (some species show a preferential occurrence for certain habitats, altitudes, or land uses) [14]. Furthermore, the genetics of conspecific AMF isolates from the same field can differ drastically [16,17], and these biological differences make it challenging to identify the prior existence of the introduced AMF in the target field. In addition, genetically divergent but closely related AMF isolates can undergo nonself fusion, giving rise to distinct genotypes that further complicates the assessment of AMF community diversity and functions [18]. Although isolate-specific markers are emerging, their application is still restricted to certain AMF species [19]. Therefore, a broader view on invasion must reflect this reality. AMF invasion is better simplified by the framework of Kinnuen et al., 2016 [10], which explains that any microbial type not currently present in the resident community is a potential invader. Through this framework, isolates or strains of commercial inoculants would be seen as potential invaders, even if they were from the so-called generalist guild, i.e., *Rhizophagus irregularis*, regardless of their potential impacts on the local community. This framework complements that of Thomsen et al., 2018 [20], who relied on the framework of Blackburn et al., 2011 [21] and concluded that some commercial AMF isolates are more likely to be invasive, and that possible invasion must be curbed by preventing initial establishment. However, we adopt the perspective that all commercial AMF isolates are potential invaders, owing to the difficulty of determining the composition of the native community and whether the inoculated strains have inhabited such a community before their introduction. Therefore, AMF invasion is simplified and can be seen as a normal event occurring in all communities.

## 3. Fundamental Ecological Processes Driving Community Dynamics Post-Inoculation

Fundamental ecological processes, such as selection, drift, dispersal, and speciation [11], determine the fate and impact of invaders on the local community. The effect of selection on the invading AMF can be positive or negative depending on the presence or absence of niche overlap between the invader and indigenous species. An invading species is likely to co-exist with the resident community if there is an empty niche space. Co-existence can be facilitated by high phylogenetic distance between the invaders and indigenous communities, or by the possession of traits absent in the local community [22]. On the other hand, phylogenetic relatedness between invaders and the indigenous community will lead to competition for available resources that can jeopardize the establishment of the invading strain, or induce significant alterations in the structure and composition of the indigenous community [7,23,24,25,26]. This has been observed in AMF-inoculated fields, where the introduced AMF failed to establish in the fields with highly diverse indigenous AMF communities [23]. A commercial inoculum containing *Rhizophagus irregularis* completely suppressed indigenous *R. irregularis* and decreased the abundance of other closely related taxa belonging to the genera *Glomus* and *Funneliformis.* At the same time, the commercial inoculum increased the abundance of two distant taxa, *Claroideoglomus* and *Paraglomus* [7]. A high density of the invading AMF could also enhance success of invasion [26]. However, the effects of selection on invading species are usually temporal, as shown in many field trials where the invading AMF have been suppressed or eliminated in plant roots after long period of time [7,24,25]. The effects of selection on invading species can also be spatiotemporal due to the adaptation of the indigenous AMF community to the local conditions or coevolution with the native plant community. Moreover, impacts of AMF inoculants on the indigenous community can be minimal in ecosystems with highly diverse and functional indigenous microbial communities, but there is also a risk of failed inoculation attempts.

Vellend (2010) [11] defined drift as random changes in the abundance of species. Invasion success is high if an invader proliferates at a higher rate than the native species [10]. Conversely, introduced strains with inferior reproductive or competitive traits tend to disappear over time. Members of *Glomeraceae* are more likely to be successful invaders, followed by those belonging to *Gigasporaceae* and *Acaulosporaceae*. This is because the *Glomeraceae* taxa possess quality traits, such as a high growth rate, hyphal anastomosis formation, high turnover rate, the ability to reproduce from spores and hyphae, and a high rate of sporulation. These traits promote proliferation compared to members of the *Gigasporaceae* family, which have delayed sporulation and recovery from hyphal damage due to their large spore size. Moreover, the *Acaulosporaceae* family is characterized by limited soil colonization, slow growth, prolonged dormancy, and low spore viability [27].

Dispersal and speciation can change the trajectory of the impacts of selection and drift on the invasion scenario. Dispersal can increase the richness of local species and reduce the negative impact of drift and selection on the invader. In addition to the ability to spread via hyphae from the inoculation point to non-target soil [28], AMF have other dispersal mechanisms, such as water, wind, and micro- and macro-fauna [20]. Members of the *Glomeraceae* family possess traits that enhance their efficient dispersal. They can be easily dispersed by wind due to their smaller spore size [24] thus, are more likely to be successful invaders. However, data are generally scarce on AMF dispersal [25]. Therefore, understanding the role of dispersal in AMF invasion requires further studies. The species richness or composition of AMF in two distinct areas with similar environmental conditions but in different geographical zones is not the same [29,30,31]. Therefore, the consequences of invasion are likely to differ between two geographical sites. Thus, it is not surprising that the effect of commercial inoculants on the indigenous AMF community are usually site specific, having a low success rate in sites with highly diverse indigenous communities [7,32,33].

## 4. Successful Invasion Does Not Necessitate Negative Impacts on the Local Community

Commercial inoculants are intended to supply functions that indigenous AMF species are supposedly incapable of providing to target crops. However, this would require successful establishment and persistence in roots and soil (at least for the desired period), leading to changes in the community dynamics of the indigenous species. Moreover, strains intended for commercial inoculants are often targeted for characteristics such as a high growth rate, resource utilization efficiency, and superior competitive abilities that are essential for their successful establishment and survival in a diverse ecosystem [20,34]. Intervention with commercial inoculants could promote AMF invasion especially in degraded soils, with low local diversity of indigenous AMF, which would not necessarily produce negative consequences [35]. Instead, it could generate both negative and positive impacts [10]. Inoculation can increase [36,37], reduce [38,39], or have neutral [32,38,40,41,42,43,44] or mixed effects [7,45] on the abundance and composition of indigenous AMF colonizing the same roots.

Therefore, AMF inoculants pose little or no risk to indigenous species if they are composed of strains with a strong symbiotic interactions, but the stakes can be high if low-quality mutualist strains are applied [8]. Since a high colonization rate does not always translate to plant benefits, the prospective commercial AMF inoculants should be screened for both colonization efficiency and strong mutualistic traits. Overall, disturbed communities are likely to recover from disruption, however, it is not clear whether they would return to the initial functional state. Further investigations of community resilience will foster our understanding about the potential risks posed by inoculum introduction to native species in terms of the loss of diversity and functions.

## 5. Perspectives

The diversity of species and functions of soil microbiota are important to modern agriculture, faced with the dilemma of feeding the growing population and reducing the environmental impacts of agriculture. Bioinoculants are interesting alternatives that contribute to achieving this goal owing to the prominent role they play in the ecological fitness of crops and soil health. Inoculation with AMF has the potential to promote crop yield and quality, and to protect crops against abiotic stress and pathogen attacks [46,47,48,49]. AMF also promote soil health by enhancing soil aggregate formation through the production of glomalin-related proteins, facilitating the biogeochemical cycle of soil minerals, controlling the turnover of soil organic matter, and promoting carbon sequestration [50,51,52]. Therefore, improving the diversity and functions of AMF communities will contribute to the sustainability of the agroecosystem now and in the future. However, it is important to bear in mind that, despite the potential benefits they can provide, foreign inoculants pose a significant threat, at least in the short term, to the resident microbial community.

Farm management practices affect the composition and structure of indigenous AMF communities, although the intensity of impacts can differ depending on AMF species and isolates [27,53]. Conservation-based agriculture adopting no-tillage and cover cropping may not require inoculation if such sites are rich in indigenous AMF communities. On the other hand, intensively managed agricultural fields usually have a low abundance of indigenous species due to mechanical disturbance and excessive chemical application; thus, such fields are likely to benefit from AMF inoculation but can also be at a disadvantage, considering the negative consequences of microbial invasion. Therefore, efforts should focus on reducing the potential risks that may result from applying such inoculants. In lieu of this, we provide our recommendations below.

### 5.1. Adopting Practices That Promote the Richness and Diversity of Indigenous Microbiota

The richness of arbuscular mycorrhizal fungal species is affected by crop species, management practices, and other environmental factors in the agroecosystem [14,54,55,56]. In undisturbed soil, i.e., no-till cropping, cover crops or crop rotation could benefit succeeding main crops by supplying AMF communities [47,52,53]. Thus, designing a mycorrhiza-friendly cropping system using a diverse rotational sequence, utilizing cover crops, and adopting no-till agriculture [57] and an agroecological approach, such as a donor crop, where the donor crop is intercropped with the target crop, could help harness the functional diversity of indigenous AMF communities [58]. Moreover, inoculation via donor crop is a more convenient and effective inoculation approach in large-field or woody crops where inoculation could be labor intensive or technically difficult [36]. However, donor crops must be carefully selected, as there could be asymmetry in the benefits gained from common mycorrhizal networks, in which the receiver plant’s investment in these networks is greater than the gains [59].

### 5.2. Inoculation with Native Propagules

Inoculation may be necessary to enhance indigenous community density in some environments, such as reclaimed mining sites, polluted sites, or disturbed agricultural fields. In this case, local propagules, if available, should be given priority.

In addition to assuring the protection of local biodiversity conferred by lower risks of invasion due to greater ecological similarity between new and existing species [11], native AMF are effective inoculants, sometimes outperforming foreign strains because they are ecologically and genetically more adaptable to the local environment [60]. Davidson et al., 2016 [61] reported that inoculation with native AMF isolates increased colonization and survival during summer, and reduced seedling mortality, indicating the potential benefit and lower risk when native species are utilized. Other reports have also confirmed the effectiveness of native AMF compared to exogenous species [56,62]. Native strains proved to be effective in improving the mycorrhization and yield of target crops in different trials in West Africa [63,64]. The geographic range for defining native AMF remains to be clarified, as reported by Michalis et al., 2012 [65], which indicates that inoculation with some native strains could also affect the diversity of other root colonizers.

Nevertheless, to avoid the deliberate introduction of invasive species, we encourage the application of native AMF to augment the background level of AMF since they are already adapted to local edaphic and climatic conditions and are likely to survive and propagate better than foreign species. Field-sourced native AMF had stronger effects than commercial AMF species [60]. Thus, the development of effective local propagules should be encouraged in the local industry. However, effective measures will be needed to monitor the movement of inoculants to unintended regions. Moreover, regulation and quality control measures are also important to assess the effectiveness of such strains.

### 5.3. Regulation and Quality Control of Commercial AMF Inoculants

Native AMF species may fail to provide the desired benefits under local conditions [23,66], necessitating intervention with commercial AMF. Moreover, indigenous AMF will most likely benefit crops in undisturbed soil. However, intensively managed agricultural fields or degraded soils will rely on external AMF input due to the disruption of the indigenous communities [67]. Therefore, the bioinoculant industry remains important to sustainable agriculture, especially with the increasing human pressure [68] for food despite unprecedented climate extremes. However, an effective system must be put in place to ensure quality and effective bioinoculants, and to avoid unintended consequences on the ecosystem.

Inoculant attributes such as the number, quality, and viability of propagules; the infectivity of strains; the composition of the inoculant, whether single or consortia of AMF species or combined with other microbes; the type of carrier and additives; and dosage recommendation [34] are crucial to the success of commercial AMF inoculants in the field and must be properly controlled and regulated. Cost-effective testing must be performed on inoculants to avoid potential contaminations and erroneous claims. Manufacturers should also administer an economically and ecologically effective dosage and provide recommendations on conditions warranting re-inoculation after the first application. Knowing the mycorrhizal potential of the target site is also crucial to making decisions on investment in commercial inoculation.

Determining the invasiveness of species requires knowledge about the past and present AMF diversity and community structure at a given site [12]. TREE (testing, regulation and monitoring, engineering, and eradication, recommended by Jack et al., 2021 [5]) remains irreplaceable to protect biodiversity, as well as plants and animals, from the negative impacts of invasion. However, regulatory institutions must be empowered, especially in developing countries (in Asia and Africa), to carry out such missions effectively and efficiently. Quantitative real-time PCR and next-generation sequencing, which rely on isolate specific probes, offer valuable resources for monitoring the persistence of AMF installation and the quality control of large-scale inoculum production [69,70]. Such tools could also be deployed in field trials, especially where the desired benefits have declined post-inoculation compared to the control. This will facilitate an understanding of the underlying factors behind plant responses. It can also help to determine if such a decline in target response is due to an alteration in the composition of native communities and how long the effect will last.

## Data Availability

Not applicable.

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
