# Peer review of "Does Commercial Inoculation Promote Arbuscular Mycorrhizal Fungi Invasion?"

_microorganisms, 2022, doi:10.3390/microorganisms10020404_

Round 1

Reviewer 1 Report

I find it very appropriate to address this issue. However, I believe that it must be taken into consideration that the introduction of new crops or the opening of farmland will change the dynamics of the microbial community as the effect on the microbiota is important. Given the variability of responses reported in the literature, wouldn't it be necessary to consider the crops to which biofertilizers will be applied for regulation?

Author Response

Response to the reviewer's comments

We are pleased to hear that our manuscript is appropriate and we thank the reviewer for these comments which were pertinent and helpful. We agree that introduction of new crops or the opening of farmland is an interesting strategy that of increases diversification rotation systems that could result on changing the dynamics of the microbial community as the effect on the microbiota. There are many factors that can change community structure of AMF communities, one of them is crop identity. Crops can also have its own microbiome which can be disturbed upon inoculation. Moreover, some crops or cultivar respond well to inoculation than other (high mycorhizotrophic crops such as leek, sorghum, corn; versus low mycorhizotrohic crops such as wheat), however, there’s no explicit evidence on AMF specificity crops. This issue was not addressed in this perspective manuscript since it focuses on AMF inoculant’s invasion risks.

Reviewer 2 Report

Many original studies and reviews have been published on the negative/positive/neutral impacts of AMF inoculants. Although this perspective manuscript does not provide a new and original perspective on them, it is a useful opinion because it includes the ecological and experimental studies and describes the future prospects for agriculture.

I think the present form is sufficient, but it is difficult to see how the diversity of AMF species and functions can lead to the sustainability and enhancement of agriculture. If the authors emphasize the diversity of species and functionality, it would help the reader's understanding if they could give some more concrete examples of why this could be justified.

The significance of AMF inoculants differs considerably between large-scale, uniform, machine-based agriculture and small-scale, soil conservation agriculture. It would be helpful to mention what kind of cultivation system will require AMF inoculation in the future.

Author Response

We thank the reviewer for these comments which were helpful for making our revision. Changes in the revised manuscript are highlighted in red.

Comment: I think the present form is sufficient, but it is difficult to see how the diversity of AMF species and functions can lead to the sustainability and enhancement of agriculture. If the authors emphasize the diversity of species and functionality, it would help the reader's understanding if they could give some more concrete examples of why this could be justified.

Response:   We address the issue raised here in line 103-110 where we mentioned how diversity of indigenous species could help prevent invasion of an exotic species, and 154-169, by highlighting importance of AMF to crop and soil health.

Comment: The significance of AMF inoculants differs considerably between large-scale, uniform, machine-based agriculture and small-scale, soil conservation agriculture. It would be helpful to mention what kind of cultivation system will require AMF inoculation in the future.

Response: We Addressed this in line 168-177 by highlighting the differences in conservation-based agriculture and intensively management agricultural systems regarding risks of invasion.